# AN EFFICIENT ENCODER-DECODER ARCHITECTURE WITH TOP-DOWN ATTENTION FOR SPEECH SEPARATION

**Kai Li[1], Runxuan Yang[1] & Xiaolin Hu[1,2,3,*]**
1. Department of Computer Science and Technology, Institute for AI,
BNRist, Tsinghua University, Beijing 100084, China
2. Tsinghua Laboratory of Brain and Intelligence (THBI),
IDG/McGovern Institute for Brain Research, Tsinghua University, Beijing 100084, China
3. Chinese Institute for Brain Research (CIBR), Beijing 100010, China
{lk21,yangrx20}@mails.tsinghua.edu.cn
xlhu@tsinghua.edu.cn

## ABSTRACT

Deep neural networks have shown excellent prospects in speech separation tasks. However, obtaining good results while keeping a low model complexity remains challenging in real-world applications. In this paper, we provide a bio-inspired efficient encoder-decoder architecture by mimicking the brain's top-down attention, called TDANet, with decreased model complexity without sacrificing performance. The top-down attention in TDANet is extracted by the global attention (GA) module and the cascaded local attention (LA) layers. The GA module takes multi-scale acoustic features as input to extract global attention signal, which then modulates features of different scales by direct top-down connections. The LA layers use features of adjacent layers as input to extract the local attention signal, which is used to modulate the lateral input in a top-down manner. On three benchmark datasets, TDANet consistently achieved competitive separation performance to previous state-of-the-art (SOTA) methods with higher efficiency. Specifically, TDANet's multiply-accumulate operations (MACs) are only 5% of Sepformer, one of the previous SOTA models, and CPU inference time is only 10% of Sepformer. In addition, a large-size version of TDANet obtained SOTA results on three datasets, with MACs still only 10% of Sepformer and the CPU inference time only 24% of Sepformer. Our study suggests that top-down attention can be a more efficient strategy for speech separation.

## 1 INTRODUCTION

In cocktail parties, people's communications are inevitably disturbed by various sounds (Bronkhorst, 2015; Cherry, 1953), such as environmental noise and extraneous audio signals, potentially affecting the quality of communication. Humans can effortlessly perceive the speech signal of a target speaker in a cocktail party to improve the accuracy of speech recognition (Haykin & Chen, 2005). In speech processing field, the corresponding challenge is to separate different speakers' audios from the mixture audio, known as *speech separation*.

Due to rapid development of deep neural networks (DNNs), DNN-based speech separation methods have significantly improved (Luo & Mesgarani, 2019; Luo et al., 2020; Tzinis et al., 2020; Chen et al., 2020; Subakan et al., 2021; Hu et al., 2021; Li & Luo, 2022). As in natural language processing, the SOTA speech separation methods are now embracing increasingly complex models to achieve better separation performance, such as DPTNet (Chen et al., 2020) and Sepformer (Subakan et al., 2021). These models typically use multiple transformer layers (Vaswani et al., 2017) to capture longer contextual information, leading to a large number of parameters and high computational

---
*Corresponding author.

cost and having a hard time deploying to edge devices. We question whether such complexity is always needed in order to improve the separation performance.

Human brain has the ability to process large amounts of sensory information with extremely low energy consumption (Attwell & Laughlin, 2001; Howarth et al., 2012). We therefore resort to our brain for inspiration. Numerous neuroscience studies have suggested that in solving the cocktail part problem, the brain relies on a cognitive process called *top-down attention* (Wood & Cowan, 1995; Haykin & Chen, 2005; Fernández et al., 2015). It enables human to focus on task-relevant stimuli and ignore irrelevant distractions. Specifically, top-down attention modulates (enhance or inhibit) cortical sensory responses to different sensory information (Gazzaley et al., 2005; Johnson & Zatorre, 2005). With neural modulation, the brain is able to focus on speech of interest and ignore others in a multi-speaker scenario (Mesgarani & Chang, 2012).

We note that encoder-decoder speech separation networks (e.g., SuDORM-RF (Tzinis et al., 2020) and A-FRCNN (Hu et al., 2021)) contain top-down, bottom-up, and lateral connections, similar to the brain's hierarchical structure for processing sensory information (Park & Friston, 2013). These models mainly simulate the interaction between lower (e.g., A1) and higher (e.g., A2) sensory areas in primates, neglecting the role of higher cortical areas such as frontal cortex and occipital cortex in accomplishing challenging auditory tasks like the cocktail party problem (Bareham et al., 2018; Cohen et al., 2005). But they provide good frameworks for applying top-down attention mechanisms.

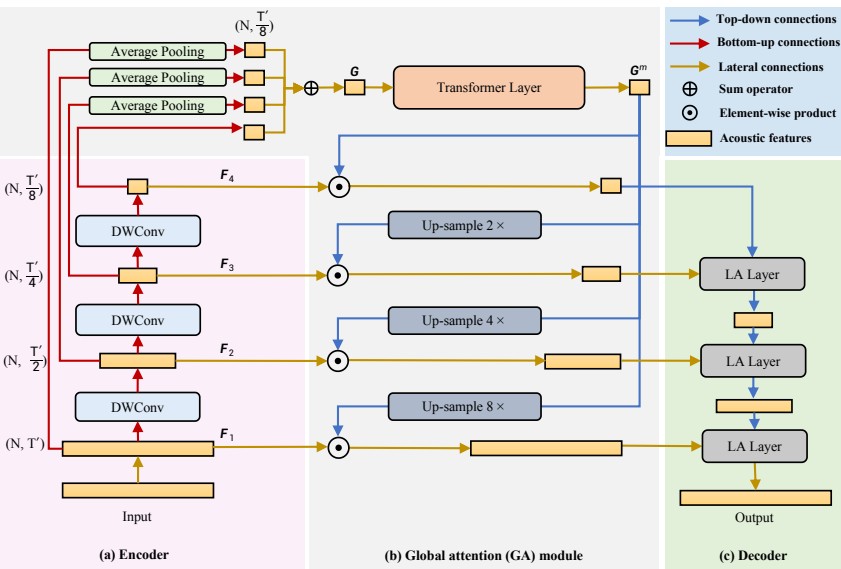

Figure 1: Main architecture of TDANet. $N$ and $T'$ denote the number of channels and length of features, respectively. By down-sampling $S$ times, TDANet contains $S + 1$ features with different temporal resolutions. Here, we set $S$ to 3. The red, blue, and orange arrows indicate bottom-up, top-down, and lateral connections, respectively. (a) The structure of the encoder, where "DWConv" denotes a depthwise convolutional layer with a kernel size of 5 and stride size of 2 followed by GLN. (b) The "Up-sample" layer denotes nearest neighbor interpolation. (c) The structure of decoder, where the LA layer adaptively modulates features of different scales by a set of learnable parameters.

In the speech separation process, the encoder and decoder information are not always useful, so we need an automatic method to modulate the features transmitted by the lateral and top-down connections. We propose an encoder-decoder architecture equipped with top-down attention for speech separation, which is called TDANet. As shown in Figure 1, TDANet adds a global attention (GA) module to the encoder-decoder architecture, which modulates features of different scales in the encoder top-down through the attention signal obtained from multi-scale features. The modulated features are gradually restored to high-resolution auditory features through local attention (LA) layers in the top-down decoder. The experimental results demonstrated that TDANet achieved competitive separation performance on three datasets (LRS2-2Mix, Libri2Mix (Cosentino et al., 2020), and WHAM! (Wichern et al., 2019)) with far less computational cost. Taking LRS2-2Mix dataset as an

example – when comparing with Sepformer, TDANet's MACs are only 5% of it and CPU inference time is only 10% of it.

## 2 RELATED WORK

In recent years, DNN-based speech separation methods have received widespread attention, and speech separation performance has been substantially improved. Existing DNN-based methods are categorized into two major groups: time-frequency domain (Hershey et al., 2016; Chen et al., 2017; Yu et al., 2017) and time domain (Luo & Mesgarani, 2019; Luo et al., 2020; Tzinis et al., 2020; Chen et al., 2020; Subakan et al., 2021; Hu et al., 2021). Time domain methods obtain better results compared to time-frequency domain methods as they avoid explicit phase estimation. Recently, time domain methods Sepformer (Subakan et al., 2021) and DPTNet (Chen et al., 2020) replace LSTM layers with transformer layers to avoid performance degradation due to long-term dependencies and to process data in parallel for efficiency. These methods achieve better separation performance, but the number of model parameters and computational cost become larger due to additional transformer layers. Although large-scale neural networks can achieve better performance in various tasks such as BERT (Devlin et al., 2019) and DALL·E 2 (Ramesh et al., 2022), it is also an important topic to design lightweight models that can be deployed to low-resource platforms.

The encoder-decoder speech separation model SuDORM-RF (Tzinis et al., 2020) achieves a certain extent of trade-off between model peformance and complexity, but its performance is still far from SOTA methods. Other researchers have designed a fully recurrent convolutional neural network A-FRCNN (Hu et al., 2021) with an asynchronous update scheme and obtained competitive separation results with a relatively small number of parameters, but the computational complexity still has room to improve.

Some earlier works (Xu et al., 2018; Shi et al., 2018; 2019) make use of top-down attention in designing speech separation models, but they only consider the projection of attentional signals to the top layer of the multilayer LSTM and without applying attention to lower layers. These approaches may not fully take advantage of top-down attention. In addition, these approaches have a large gap in model performance and complexity with recent models including SuDORM-RF and A-FRCNN.

For a survey on the use of top-down attention in other domains, see Appendix A

## 3 METHOD

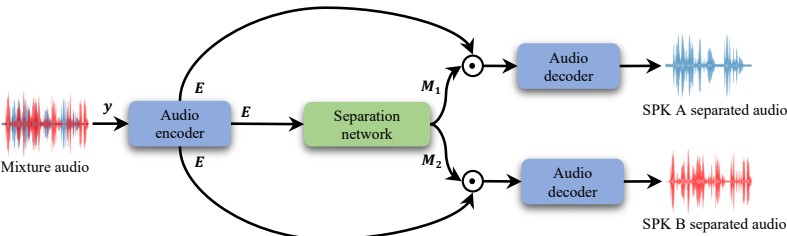

Figure 2: The overall pipeline for the speech separation task. We assume that the mixture audio contains two speakers here. The parameters are shared between the two audio decoders.

### 3.1 OVERALL PIPELINE

Given an audio containing multiple speakers, the speech separation methods aim to extract different speakers' utterances and route them to different output channels. Let $y \in \mathbb{R}^{1 \times T}$ be a multi-speaker time-domain audio signal with the length $T$:

$$y = \sum_{i}^{C} x_i + n,$$

(1)

consisting of $C$ speaker signals $\boldsymbol{x}_i \in \mathbb{R}^{1 \times T}$ plus the noise signal $\boldsymbol{n} \in \mathbb{R}^{1 \times T}$. We expect the separated speeches $\bar{\boldsymbol{x}}_i \in \mathbb{R}^{1 \times T}$ from all speakers to be closer to target speeches $\boldsymbol{x}_i$.

Our proposed method uses the same three-part pipeline as in Conv-TasNet (Luo & Mesgarani, 2019): *an audio encoder*, *a separation network* and *an audio decoder*. The audio encoder transforms $\boldsymbol{y}$ into a frame-based $N$-dimensional embedding sequence $\boldsymbol{E} \in \mathbb{R}^{N \times T'}$ with length $T'$, called mixture audio embedding, which can be implemented using a 1-D convolutional layer with kernel size $L$ and stride size $\lfloor L/4 \rfloor$. Similar to existing separation methods (Luo & Mesgarani, 2019; 2018; Luo et al., 2020; Subakan et al., 2021; Chen et al., 2020; Li et al., 2022a), instead of directly estimating the target speech embeddings, we use a DNN-based separation network $f(\boldsymbol{E}; \boldsymbol{\theta})$ to generate a set of masks $\boldsymbol{M} = \{\boldsymbol{M}_i \in \mathbb{R}^{N \times T'} | i = 1, ..., C\}$ associated with the target speakers. Each target speech embedding $\bar{\boldsymbol{E}}_i \in \mathbb{R}^{N \times T'}$ is generated by applying the corresponding mask $\boldsymbol{M}_i$ to the mixture audio embedding $\boldsymbol{E}$:

$$\bar{\boldsymbol{E}}_i = \boldsymbol{E} \odot \boldsymbol{M}_i, \tag{2}$$

where $\odot$ denotes element-wise product. Finally, the target waveform $\bar{\boldsymbol{x}}_i$ is reconstructed using the target speech embedding $\bar{\boldsymbol{E}}_i$ through an audio decoder, which can be implemented using a 1-D transposed convolutional layer with the same kernel and stride sizes as the audio encoder.

Overall there are three main components: (a) encoder (see Section 3.2), (b) global attention (GA) module (see Section 3.3) and (c) decoder (see Section 3.4). The process detail is as follows. First, the encoder obtains multi-scale features using the bottom-up connections. Second, the GA module extracts global features at the top layer using multi-scale features, and then the global features are used as attention to modulate features from different scales using top-down connections. Third, the adjacent layer features are used as the input into the decoder to extract the acoustic features through LA layers. This completes one cycle of information processing inside TDANet. Similar to A-FRCNN Hu et al. (2021), we cycle TDANet several times and use the output features from the last cycle as the separation network output (see Section 3.5).

## 3.2 ENCODER OF TDANET

The encoder of TDANet is designed for extracting features at different temporal resolutions. Lower layers have higher temporal resolutions, while higher layers have lower temporal resolutions. The encoder works as follows (see Figure 1a). For a mixture audio embedding $\boldsymbol{E}$, the encoder processes $\boldsymbol{E}$ in a bottom-up manner step by step. The bottom-up connections in the encoder are implemented by down-sampling layers, which consist of a 1-D convolutional layer followed by a global layer normalization (GLN) (Luo & Mesgarani, 2019) and PReLU. To efficiently expand the perceptual field, these convolutional layers use dilation convolution with $N$ kernels with size 5, stride size 2 and dilation size 2 instead of standard convolution to aggregate longer contexts. In this way, we obtain features with different resolutions $\{\boldsymbol{F}_i \in \mathbb{R}^{N \times \frac{T'}{2^{i-1}}} | i = 1, ..., S+1\}$, where $S$ denotes the number of down-sampling.

## 3.3 GLOBAL ATTENTION (GA) MODULE OF TDANET

The GA module works in two steps:

(1) It receives multi-scale features $\{\boldsymbol{F}_i | i = 1, ..., S+1\}$ as input and computes global feature $\mathbf{G}^m \in \mathbb{R}^{N \times \frac{T'}{2^S}}$;

(2) It uses the $\mathbf{G}^m$ as top-down attention to modulate $\{\boldsymbol{F}_i | i = 1, ..., S+1\}$, controlling the redundancy of information passed to the decoder.

**Details of the first step in the GA module**: Taking multi-scale features $\{\boldsymbol{F}_i | i = 1, ..., S+1\}$ as input, we use the average pooling layers to compress these features in temporal dimension from $\frac{T'}{2^{i-1}}$ to $\frac{T'}{2^S}$ for reducing computational cost. The features with different temporal resolutions are fused into the global feature $\mathbf{G} \in \mathbb{R}^{N \times \frac{T'}{2^S}}$ by a summation operation. The calculation of global feature can be written as:

$$\mathbf{G} = \sum_i^S p(\boldsymbol{F}_i), \tag{3}$$

where $p(\cdot)$ denotes the average pooling layer. $\mathbf{G}$ will be treated as input to the transformer layer to obtain the speaker's acoustic patterns from global contexts.

The transformer layer contains two components: multi-head self-attention (MHSA) and feed-forward network (FFN) layers, as shown in Figure 3. The MHSA layer is the same as the encoder structure defined in Transformer (Vaswani et al., 2017). This layer has been widely used in recent speech processing tasks (Chen et al., 2020; Subakan et al., 2021; Lam et al., 2021). First, the position information $\boldsymbol{d}$ of different frames is added to global feature $\mathbf{G}$ to get $\bar{\mathbf{G}} = \mathbf{G} + \mathbf{d}$, $\bar{\mathbf{G}} \in \mathbb{R}^{N \times \frac{T'}{2S}}$. Then, each attention head calculates the dot product attention between all elements of $\bar{\mathbf{G}}$ in different feature spaces to direct model focus to different aspects of information. Finally, the output $\hat{\mathbf{G}} \in \mathbb{R}^{N \times \frac{T'}{2S}}$ of MHSA is fed into GLN and then connected $\mathbf{G}$ through the residual connection to obtain $\check{\mathbf{G}} \in \mathbb{R}^{N \times \frac{T'}{2S}}$.

The FFN layer followed by MHSA layer consists of three convolutional layers. First, $\check{\mathbf{G}}$ is processed in a $1 \times 1$ convolutional layer with GLN mapped to the $2N$-dimensional representation space. Then, the second convolutional layer composed of a $5 \times 5$ depthwise convolutional layer

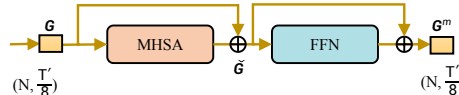

Figure 3: The structure of transformer layer.

followed by GLN extracts relationship among $2N$ sequences. Finally, a $1 \times 1$ convolutional layer followed by GLN compresses the $2N$-dimensional features into $N$-dimensions to obtain $\mathbf{G}^m \in \mathbb{R}^{N \times \frac{T'}{2S}}$ and also use a residual connection to alleviate the vanishing gradient problem.

**Details of the second step in the GA module**: We use $\mathbf{G}^m$ as top-down attention (see Figure 1b) to modulate $\boldsymbol{F}_i$ before adding it to the decoder. Specifically, we up-sample $\mathbf{G}^m$ along the time dimension using nearest neighbor interpolation to obtain the same time dimension as $\boldsymbol{F}_i$, and then use the Sigmoid function to obtain the attention signal. In this way, the multi-scale semantic information of $\mathbf{G}^m$ can guide the local features $\boldsymbol{F}_i$ to focus more on feature details that may be lost in the bottom-up path in order to improve quality of separated audios. The modulated features are calculated as:

$$\boldsymbol{F}_i' = \sigma(\phi(\mathbf{G}^m)) \odot \boldsymbol{F}, \qquad (4)$$

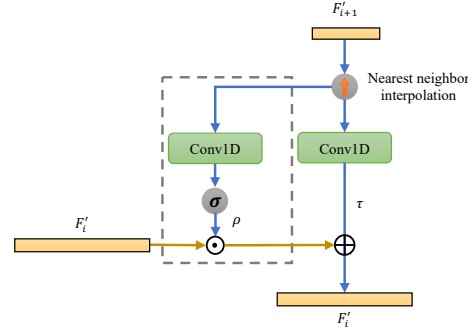

Figure 4: The structure of LA layer in the decoder, where "Conv1D" denotes the $5 \times 5$ depthwise convolutional layer followed by GLN.

where $\phi$ denotes nearest neighbor interpolation, $\sigma$ stands for the Sigmoid function and $\odot$ denotes element-wise product. The modulated features $\boldsymbol{F}_i' \in \mathbb{R}^{N \times \frac{T'}{2^{i-1}}}$ are used as input to the decoder to extract different speakers' features.

The GA module functions as higher cortical areas such as frontal cortex and occipital cortex in the brain which play an important role in accomplishing complicated sensory tasks. Here, it is only designed for top-down modulation in performing speech separation tasks (Bareham et al., 2018; Cohen et al., 2005).

## 3.4 DECODER OF TDANET

The decoder progressively extracts acoustic features through top-down connections whose core component is the LA layer, as shown in Figure 1c. The LA layer uses a small amount of parameters ($\sim$ 0.01M) to generate adaptive parameters $\rho$ and $\tau$ as local top-down attention to perform affine transformation on features from the current layer to reconstruct fine-grained features. Figure 4 shows detailed architecture of the LA layer. First, LA layer uses $N$ 1-D kernels with length 5 to perform convolution in temporal dimension using features $\boldsymbol{F}_{i+1}'$, resulting in $\tau \in \mathbb{R}^{N \times \frac{T'}{2^{i-1}}}$. Meanwhile, an-

other identical 1-D convolutional layer followed by GLN and Sigmoid uses $\boldsymbol{F}'_{i+1}$ as input, resulting in attention signal $\rho \in \mathbb{R}^{N \times \frac{T'}{2^{i-1}}}$. Consequently,

$$\tau = h_2(\phi(\boldsymbol{F}'_{i+1})), \rho = \sigma(h_1(\phi(\boldsymbol{F}'_{i+1}))), \tag{5}$$

where $h_1$ and $h_2$ denote two different parameters of 1-D convolutional layers followed by GLN. $\boldsymbol{F}'_i$, $\rho$ and $\tau$ have the same dimensions. The learnable parameters are used to adaptively modulate $\boldsymbol{F}'_i$ via a top-down local attention $\rho$. This process is formulated by

$$\boldsymbol{F}'_i = \rho \odot \boldsymbol{F}'_i + \tau. \tag{6}$$

After the feature extraction from the highest layer to the lowest layer, the output of the decoder use $C$ convolutional layers with $N$ kernels with length of 1 and stride of 1 followed by an activation function (ReLU) to obtain $C$ masks $\boldsymbol{M}_i$, respectively.

## 3.5 UNFOLDING METHOD OF TDANET

Clearly, the bottom-up, lateral and top-down connections (Figure 1) make TDANet a recurrent model. We need to unfold it through time for training and testing. We adopt the unfolding scheme summation connection (SC) as in A-FRCNN (Hu et al., 2021). Figure 1 shows a single block in the whole unfolding scheme. We repeat the block $B$ times (weight sharing), such that the model's input adds up with each block's output as the next block's input.

## 4 EXPERIMENT CONFIGURATIONS

### 4.1 DATASET SIMULATIONS

We evaluated TDANet and other existing methods on three datasets: LibriMix, WHAM!, LRS2-2Mix). The speaker identities of `training/validation` set and `test` set are non-intersecting, meaning unseen speaker data from the training phase were used during testing.

**Libri2Mix** (Cosentino et al., 2020). In this dataset, the target speech in each mixture audio was randomly selected from a subset of LibriSpeech's train-100 (Panayotov et al., 2015) and mixed with uniformly sampled Loudness Units relative to Full Scale (LUFS) (Series, 2011) between -25 and -33 dB. Each mixture audio contains two different speakers and have a duration of 3 seconds with 8 kHz sample rate.

**WHAM!** (Wichern et al., 2019). This dataset acts as a noisy version of WSJ0-2Mix (Hershey et al., 2016). In the WSJ0-2Mix dataset, the overlap between two speaker audios in the mixture is 100%, which is too idealistic, and thus model trained on this dataset does not achieve generalization to speeches from a broader range of speakers (Cosentino et al., 2020). In the WHAM! dataset, speeches were mixed with noise recorded in scenes such as cafes, restaurants and bars. The signal-to-noise ratio of the noise was uniformly sampled between -6db and 3db, making it more challenging mixture audios than WSJ0-2Mix. Each mixture audio contains two different speakers and have a duration of 4 seconds with 8 kHz sample rate.

**LRS2-2Mix**. The LRS2 dataset (Afouras et al., 2018) contains thousands of video clips acquired through BBC. LRS2 contains a large amount of noise and reverberation interference, which is more challenging and closer to the actual environment than the WSJ0 (Garofolo et al., 1993) and LibriSpeech (Panayotov et al., 2015) corpora. We created a new speech separation dataset, LRS2-2Mix, using the LRS2 corpus, where the training set, validation set and test set contain 20000, 5000 and 3000 utterances, respectively. The two different speaker audios from different scenes with 16 kHz sample rate were randomly selected from the LRS2 corpus and were mixed with signal-to-noise ratios sampled between -5 dB and 5 dB. The data were simulated using a script consistent with WSJ0-2Mix[1]. The length of mixture audios is 2 seconds. It is publicly available[2].

---

[1] `http://www.merl.com/demos/deep-clustering/create-speaker-mixtures.zip`
[2] `https://drive.google.com/file/d/1dCWD5OIGcj43qTidmU18unoaqo_6QetW/view`

## 4.2 MODEL AND TRAINING CONFIGURATIONS

We set the kernel size $L$ of the *audio encoder* and *audio decoder* in the overall pipeline to 4 ms and stride size $\lfloor L/4 \rfloor$ to 1 ms. The number of down-sampling $S$ was set to 4. The number of channels $N$ of all convolutional layers in each layer was set to 512, and the SC method proposed by A-FRCNN (Hu et al., 2021) was used to unfold $B = 16$ times at the macro level. For the MHSA layer, we set the number of attention heads to 8, the dimension to 512, and used sine and cosine functions with different frequencies for the position encoding. The mask of the MHSA layer was disabled because our task allows the model to see future information. In addition, we set the number of channels for the three convolutional layers in the FFN layer to (512, 1024, 512), the kernel sizes to (1, 5, 1), the stride sizes to (1, 1, 1), and the bias settings to (False, True, False). To avoid overfitting, we set the probability of all dropouts to 0.1.

We trained all models for 500 epochs. The batch size was set to 1 at the utterance level. Our proposed model used the Adam (Kingma & Ba, 2015) optimizer with an initial learning rate of 0.001. We used maximization of scale-invariant signal-to-noise ratio (SI-SNR) (Hershey et al., 2016) as the training goal (See Appendix B for details). Once the best model was not found for 15 successive epochs, we adjusted the learning rate to half of the previous one. Moreover, we stopped training early when the best model was not found for 30 successive epochs. During the training process, we used gradient clipping with a maximum L2 norm of 5 to avoid gradient explosion. For all experiments, we used $8\times$ GeForce RTX 3080 for training and testing. The PyTorch implementation of our method is publicly available[3]. This project is under the MIT license.

## 4.3 EVALUATION METRICS

In all experiments, we report the results of scale-invariant signal-to-noise ratio improvement (SI-SNRi) (Le Roux et al., 2019) and signal-to-distortion ratio improvement (SDRi) (Vincent et al., 2006) used to measure clarity of separated audios. For measuring model efficiency, we report the processing time consumption per second (real-time factor, RTF) for all models, indicated by "RTF" in the tables throughout this paper. It was calculated by processing ten audio tracks of 1 second in length and 16 kHz in sample rate on CPU and GPU (total processing time / 10), represented as "CPU RTF" and "GPU RTF", respectively. The numbers were then averaged after running 1000 times. Also, we used the parameter size and the number of MACs to measure the model size, where the MACs were calculated using the open-source tool PyTorch-OpCounter[4]. This toolkit is used under the MIT license.

# 5 RESULTS

## 5.1 ABLATION STUDY

To better understand the effectiveness of top-down attention, we investigated the impacts of two types of attention components in the model for separation performance, including GA module and LA layer. All experimental results were obtained by training and testing on the LRS2-2Mix dataset.

Table 1: Comparison over separation performance and efficiency among different types of top-down attention. "$\sqrt{}$" indicates the case where a component is used in the TDANet. "$\times$" indicates the opposite.

| GA module | LA layer | SI-SNRi (dB) | SDRi (dB) | Params (M) | MACs (G/s) | CPU RTF (s) |
|:---:|:---:|:---:|:---:|:---:|:---:|:---:|
| $\times$ | $\times$ | 10.1 | 10.5 | **0.7** | **2.5** | **0.65** |
| $\sqrt{}$ | $\times$ | 12.4 | 12.7 | 2.3 | 4.6 | 0.79 |
| $\times$ | $\sqrt{}$ | 11.9 | 12.2 | **0.7** | 2.6 | 0.65 |
| $\sqrt{}$ | $\sqrt{}$ | **13.2** | **13.5** | 2.3 | 4.7 | 0.79 |

We first constructed a control model that does not contain the GA module and LA layer. It takes the top-layer features of the encoder as input to the decoder. Note that excluding the LA layer means

---

[3]`https://cslikai.cn/project/TDANet/`
[4]`https://github.com/Lyken17/pytorch-OpCounter`

removing the gray box in Figure 4. This architecture then becomes a typical U-Net (Ronneberger et al., 2015), similar to a basic block used as in SuDORM-RF (Tzinis et al., 2020). From Table 1, it is seen that SI-SNRi of this model is only 10.1 dB.

When only adding the GA module into TDANet, the SI-SNRi increased by 2.3 dB compared to when it was absent (Table 1). The GA module contains a transformer layer and top-down projections. We are interested in whether the performance improvement came from the transformer layer or the top-down projections. Table 4 from Appendix C.1 demonstrates that the performance improvement is independent of the transformer layer. When only adding the LA layer, we observed that the SI-SNRi increased by 1.8 dB without increasing computational cost. We noticed that the GA module had larger impact on performance improvement. One possible explanation is that the GA module modulates on a wide range of features, while the LA layer only focuses on adjacent ones. When both GA module and LA layer were used, the performance were the best.

More ablation studies and discussions are reported in Appendix C. Firstly, we observed that multi-scale features using summation fusion strategy as GA module input were efficient for performance improvement without significantly increasing computational cost (Appendix C.2). Secondly, TDANet obtained the best separation performance when both MHSA and FFN layers were present in the GA module (Appendix C.3). Thirdly, we found that top-down attention is essential in improving separation performance (Appendix C.4). Finally, we discussed the possible reasons for the effectiveness of LA and further verified its performance on SuDoRM-RF (Appendix C.4).

## 5.2 Comparisons with state-of-the-art methods

We conducted extensive experiments to quantitatively compare speech separation performance of our proposed TDANet with some existing speech separation models on three datasets. The results are shown in Table 2. We selected some typical models that have shown good separation performance, including BLSTM-TasNet (Luo & Mesgarani, 2018), Conv-TasNet (Luo & Mesgarani, 2019), SuDORM-RF (Tzinis et al., 2020), DualPathRNN (Luo et al., 2020), DPTNet (Chen et al., 2020), Sepformer (Subakan et al., 2021), and A-FRCNN (Hu et al., 2021). SuDORM-RF contains two variants with good performance, suffixed with 1.0x and 2.5x, indicating that these variants consist of 16 and 40 blocks, respectively. A-FRCNN has a suffix "16" denoting that the model is expanded 16 times at the macro level. LRS2-2Mix is the newly proposed speech separation dataset in this paper, so none of these models reported results on this dataset.

**Separation performance**. TDANet obtained competitive separation performance with much lower complexity than previous SOTA models on the three datasets (see Table 2), clearly demonstrating the importance of top-down attention for the encoder-decoder network. On the LRS2-2Mix dataset, TDANet lost only 0.3 dB SI-SNRi in separation performance but with only 8% of number of parameters compared to Sepformer, one of previous SOTA models. On the other two datasets, TDANet also obtained competitive results compared to previous SOTA models.

Some audio examples from LRS2-2Mix separated by different models are provided in **Supplementary Material**. In most examples, we found that separation results with TDANet sound better than those from other models.

**Separation effciency**. In addition to number of model parameters, MACs as well as training and inference time are also important indicators of model complexity, as models with a small amount of parameters may also have a large number of MACs and could be slow to train and infer. By using RTF and MACs as metrics, we evaluated complexity of pervious models (Table 3). We observed that TDANet outperformed the previous SOTA models in model complexity. For example, compared to DPTNet and Sepformer, TDANet's MACs are 5% of them, and backward time is 13% and 47% of them, respectively, significantly reducing the time consumption during training. In addition, compared with A-FRCNN and SuDORM-RF 2.5x, TDANet's GPU inference time is 19% and 18% of them, GPU training time is 47% and 37% of them, and CPU inference time is 15% and 46% of them, respectively. These results suggest that TDANet can be more easily deployed to low-resource devices.

In addition, we verified the performance of TDANet model at low latency (causal) in Appendix D, and the results show that TDANet can still obtain better performance than the lightweight models (SuDORM-RF and A-FRCNN).

Table 2: Quantitative comparison between TDANet and other existing models on three datasets (test set). "-" indicates that their results have not been reported in other papers. "*" denotes that they were not reported in the original paper but were trained and tested using the Asteroid Toolkit (Pariente et al., 2020). The results of DPTNet on Libri2Mix and WHAM! datasets were reported in (Yao et al., 2022), but not in the original paper. "†" denotes that the results of Sepformer were trained and tested using the SpeechBrain Toolkit (Ravanelli et al., 2021)

| Model [Authors, Years] | LRS2-2Mix | | Libri2Mix | | WHAM! | | Params |
| --- | --- | --- | --- | --- | --- | --- | --- |
| | SI-SNRi | SDRi | SI-SNRi | SDRi | SI-SNRi | SDRi | (M) |
| BLSTM-TasNet [Luo et al, 2018] | 6.1* | 6.8* | 7.9 | 8.7 | 9.8 | - | 23.6 |
| Conv-TasNet [Luo et al, 2019] | 10.6* | 11.0* | 12.2 | 12.7 | 12.7 | - | 5.6 |
| SuDoRM-RF1.0x [Tzinis et al, 2020] | 11.0* | 11.4* | 13.5 | 14.0 | 12.9 | 13.3 | 2.7 |
| SuDoRM-RF2.5x [Tzinis et al, 2020] | 11.3* | 11.7* | 14.0 | 14.4 | 13.7 | 14.1 | 6.4 |
| DualPathRNN [Luo et al, 2020] | 12.7* | 13.0* | 16.1 | 16.6 | 13.7 | 14.1 | 2.7 |
| DPTNet [Chen et al, 2020] | 13.3* | 13.6* | 16.7 | 17.1 | 14.9 | 15.3 | 2.7 |
| A-FRCNN-16 [Hu et al, 2021] | 13.0* | 13.3* | 16.7 | 17.2 | 14.5 | 14.8 | 6.1 |
| Sepformer [Subakan et al, 2021] | 13.5† | 13.8† | 16.5 | 17.0 | 14.4† | 15.0† | 26.0 |
| TDANet (*ours*) | 13.2 | 13.5 | 16.9 | 17.4 | 14.8 | 15.0 | 2.3 |
| TDANet Large (*ours*) | **14.2** | **14.5** | **17.4** | **17.9** | **15.2** | **15.4** | 2.3 |

**Large-size version**. For the proposed method, we also constructed a large-size version of TDANet, namely TDANet Large, by modifying the kernel length of the *audio encoder* and *audio decoder* from 4ms to 2ms and the stride size from 2ms to 0.5ms. Note that this does not change the model size while also increasing complexity (Tzinis et al., 2020; Luo et al., 2020; Bahmaninezhad et al., 2019). TDANet Large obtained SOTA separation performance without a significant increase in computational cost on all three datasets (see Table 2 and Table 3). TDANet Large can still maintain a small computational complexity compared to other existing lightweight models. For example, the inference time on the GPU is about 3 times faster, and MACs are about 14 times smaller compared to A-FRCNN. We supplement the results for other models (A-FRCNN and Sepformer) with different kernel size and stride size; see Appendix E for details.

Table 3: Comparison of inference time and MACs on the LRS2-2Mix test set. The test environment for feedforward RTF (F GPU RTF) and backward RTF (B GPU RTF) is Nvidia GeForce RTX 2080 Ti, while the test environment for CPU RTF is Intel(R) Xeon(R) Silver 4210 CPU @ 2.20GHz, single-threaded.

| Model | F GPU RTF (ms) | B GPU RTF (ms) | CPU RTF (s) | MACs (G/s) |
| --- | --- | --- | --- | --- |
| BLSTM-TasNet | 233.85 | 654.14 | 5.9 | 43.0 |
| Conv-TasNet | 15.28 | 56.91 | 0.82 | 10.2 |
| SuDoRM-RF 1.0x | 27.86 | 95.37 | 0.75 | 4.6 |
| SuDoRM-RF 2.5x | 64.70 | 228.57 | 1.73 | 10.1 |
| DualPathRNN | 88.79 | 241.54 | 8.13 | 85.3 |
| DPTNet | 103.26 | 689.06 | 10.49 | 102.5 |
| A-FRCNN-16 | 61.16 | 183.65 | 5.32 | 125.3 |
| Sepformer | 65.61 | 184.91 | 7.55 | 86.9 |
| TDANet (*ours*) | 11.76 | 86.56 | 0.79 | 4.7 |
| TDANet Large (*ours*) | 23.77 | 97.92 | 1.78 | 9.1 |

## 6 Conclusions

To fill the gap between the performance of SOTA and lightweight models, we designed an encoder-decoder speech separation network with top-down global attention and local attention, inspired by our brain's top-down attention for solving the cocktail party problem. Our extensive experimental results showed that TDANet could significantly reduce model complexity and inference speed while ensuring separation performance.

Limitations: The TDANet only reflects certain working principles of the auditory system and is not a faithful model of the auditory system. For example, it requires a bottom-up encoder and a top-down decoder, and it is unclear how such components are implemented in the brain.

ACKNOWLEDGMENTS

This work was supported by the National Key Research and Development Program of China (No. 2021ZD0200301) and the National Natural Science Foundation of China (Nos. 61836014, 62061136001, U19B2034).

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

## A    RELATED WORKS IN OTHER FIELDS

To the best of our knowledge, there is no existing method on speech separation that uses top-down attention in an encoder-decoder architecture. Thus, we investigated the use of global top-down attention (similar to top-down attention in the GA module) in hierarchical models in other domains, e.g., image segmentation Chen et al. (2016); Sinha & Dolz (2020) and image fusion Li et al. (2022b). However, these methods are all feed-forward network structures. Besides, these methods use different scale features to obtain the corresponding top-down attention respectively and use top-down attention to modulate the features in a complicated way. Furthermore, we surveyed local top-down attention approaches (similar to the LA layer) in other domains, such as SENet (Hu et al., 2018), Highway network (Srivastava et al., 2015) and GRCNN (Wang & Hu, 2021). These methods introduce the attention mechanism (gate) to achieve filtering and integration of information flow in order to obtain better performance. In this paper, we focus on designing simple neural networks by using multi-scale features as top-down attention to achieve good results with high efficiency.

## B    TRAINING METHOD

We apply the permutation-invariant training (PIT) method (Yu et al., 2017) to solve the permutation problem (Hershey et al., 2016) in order to maximize the scale-invariant signal-to-noise ratio (SI-SNR) (Le Roux et al., 2019). The SI-SNR for each speaker is defined as:

$$\text{SI-SNR}_i = 20 \times log_{10} \frac{||\boldsymbol{A}_{target}||}{||\boldsymbol{e}_{noise}||}, \tag{7}$$

where

$$\boldsymbol{A}_{target} = \frac{\langle \bar{\boldsymbol{x}}_i, \boldsymbol{x}_i \rangle \boldsymbol{x}_i}{||\boldsymbol{x}_i||_2^2}, \boldsymbol{e}_{noise} = \bar{\boldsymbol{x}}_i - \boldsymbol{A}_{target}. \tag{8}$$

In the above equation, $\langle \cdot, \cdot \rangle$ denotes the inner product and $|| \cdot ||_2^2$ denotes the $L2$-norm.

## C    ADDITIONAL EXPERIMENTS

### C.1    EFFECT OF TRANSFORMER LAYER IN THE GA MODULE

The GA module contains a transformer layer and top-down projections. We are interested in whether the performance gain is from the transformer layer or the top-down projections. For TDANet with only the GA module, we removed top-down projections from the GA module, keeping only the transformer layer. When the transformer layer is present, we take the output of it as the decoder input. When the transformer layer is absent, we use the top-layer features of the encoder as input of the decoder. Table 4 shows the experimental results. We found that the presence of the transformer layer does not significantly affect separation performance; both configurations yielded poor results compared with the configuration with top-down projections (SI-SNRi 12.4 dB, Table 1). These results further demonstrate the importance of top-down global attention.

Table 4: Effect of transformer layer. "TL" denotes transformer layer in the GA module on the LRS2-2Mix dataset.

| TL | SI-SNRi (dB) | SDRi (dB) | Params (M) | MACs (G/s) | CPU RTF (s) |
|----|--------------|-----------|------------|------------|-------------|
| $\checkmark$ | **10.3** | **10.7** | 2.3 | 4.6 | 0.77 |
| $\times$ | 10.1 | 10.5 | **0.7** | **2.5** | **0.65** |

## C.2 EFFECT OF MULTI-SCALE INPUT TO GA MODULE

To examine the influence on separation performance between two different features as input to the GA module: multi-scale fused features $\mathbf{G}$ and top-layer features $\boldsymbol{F}_{S+1}$ (ResNet-style encoder), we experimented with each of them separately. The experimental results are shown in Table 5. We chose $\mathbf{G}$ instead of $\boldsymbol{F}_{S+1}$ because the former achieved better separation performance (0.4 dB $\uparrow$) without a remarkable increase in computational cost. One possible explanation is that we added dense connections to the ResNet-style structure to project features at different temporal scales to the top layer, which is similar to DenseNet Huang et al. (2017), boosting the back-propagation of gradients and making the network easier to train. Another possible explanation is that, in a ResNet-style structure, features may lose details gradually while propagating bottom-up. Using skip connections to project to the top layer enables more efficient use of multi-scale features.

Table 5: Comparison over separation performance and model complexity between different multi-scale fused top-layer features used for input into the GA module on the LRS2-2Mix dataset.

| Methods | SI-SNRi (dB) | SDRi (dB) | Params (M) | MACs (G/s) | CPU RTF (s) |
|---------|--------------|-----------|------------|------------|-------------|
| **G** | **13.2** | **13.5** | **2.3** | **4.7** | 0.79 |
| $\boldsymbol{F}_{S+1}$ | 12.8 | 13.1 | 2.3 | 4.7 | **0.74** |

In addition, we investigated two different strategies for multi-scale feature fusion: summation and concatenation. Table 6 shows the results. Summation outperformed concatenation in terms of both separation performance and computational efficiency. Therefore, we employed summation operation as the fusion strategy.

Table 6: Comparison over separation performance and model complexity between different fusion strategies on the LRS2-2Mix dataset.

| Fusion strategy | SI-SNRi (dB) | SDRi (dB) | Params (M) | MACs (G/s) | CPU RTF (s) |
|-----------------|--------------|-----------|------------|------------|-------------|
| Summation | **13.2** | **13.5** | **2.3** | **4.7** | **0.79** |
| Concatenation | 13.0 | 13.3 | 3.6 | 6.1 | 0.83 |

## C.3 EFFECT OF TRANSFORMER LAYER COMPONENTS

The transformer layer contains two components: MHSA and FFN layers. We have investigated the effectiveness of MHSA and FFN layers on separation performance and inference efficiency in Table 7. We used the TDANet with both GA module and LA layers as the control model. When we remove MHSA and FFN layers from GA, the top-layer features of the encoder are used as top-down global attention. We observed that the these layers played a critical role in improving separation performance. When adding either MHSA or FFN layer, we observed that the impact of FFN ($\uparrow$ 1.3 dB) on performance was more significant than that of MHSA ($\uparrow$ 0.7 dB). One possible reason may be that the perceptual field of the top-layer features is already large enough so that the transformer structure, known to be good at modeling long sequences, does not improve separation performance as much.

Table 7: Comparison on the importance of MHSA and FFN layers on the LRS2-2Mix dataset.

| MHSA | FFN | SI-SNRi (dB) | SDRi (dB) | Params (M) | MACs (G/s) | CPU RTF (s) |
|------|-----|--------------|-----------|------------|------------|-------------|
| × | × | 11.2 | 11.5 | **0.2** | **2.6** | **0.68** |
| √ | × | 11.9 | 12.2 | 1.3 | 3.6 | 0.73 |
| × | √ | 12.5 | 12.8 | 1.3 | 3.7 | 0.73 |
| √ | √ | **13.2** | **13.5** | 2.3 | 4.7 | 0.79 |

### C.4 ADVANTAGES OF TOP-DOWN ATTENTION IN THE GA MODULE AND DECODER

We verify the effect of global top-down attention on SuDoRM-RF. We found that adding top-down attention to the SuDoRM-RF can also improve separation performance, as shown in Table 8.

Table 8: Comparison on the importance of top-down attention (TDA) on SuDoRM-RF on the LRS2-2Mix dataset.

| Models | SI-SNRi (dB) | SDRi (dB) | Params (M) | MACs (G/s) |
|--------|--------------|-----------|------------|------------|
| SuDoRM-RF 1.0x | 11.0 | 11.4 | 2.7 | 4.6 |
| SuDoRM-RF 1.0x + TDA | **11.9** | **12.3** | **2.7** | **4.6** |

We add LA layers to the decoder of SuDoRM-RF to verify the importance of the LA layer. We found that adding LA layers can also improve separation performance, as shown in Table 9.

Table 9: Comparison on the importance of LA layers on SuDoRM-RF on the LRS2-2Mix dataset.

| Models | SI-SNRi (dB) | SDRi (dB) | Params (M) | MACs (G/s) |
|--------|--------------|-----------|------------|------------|
| SuDoRM-RF 1.0x | 11.0 | 11.4 | 2.7 | 4.6 |
| SuDoRM-RF 1.0x + LA | 11.7 | 12.0 | 3.3 | 4.9 |

## D TDANET IN LOW LATENCY CASE

For the real-time problem, we tried to modify SuDoRM-RF (Tzinis et al., 2020), A-FRCNN (Hu et al., 2021) and TDANet to causal versions. We replaced the standard convolutional layers in both models with causal convolutional layers to mask out future information (modified from SuDoRM-RF[5]). We supplemented the experiments with SuDoRM-RF (causal), A-FRCNN-16 (causal) and TDANet (causal). The results are shown in Table 10. The results show that modifying the models to causal versions degrade the separation performance to varying degrees (compare Table 2 and Table 10). However, TDANet is still able to obtain better performance than SuDoRM-RF and A-FRCNN.

## E DIFFERENT SETTINGS OF THE HYPERPARAMETER $L$

For the hyperparameter $L$, different models have different settings in their original papers. We compare the results of TDANet and the SOTA model Sepformer and lightweight model A-FRCNN with different values of $L$. It is seen from the Table 11 that, for any model, when $L$ is larger, the inference speed is faster, but the separation performance goes down. In addition, for any $L$ value, our proposed TDANet achieves the best results with the highest efficiency.

---

[5]`https://github.com/etzinis/sudo_rm_rf/blob/master/sudo_rm_rf/dnn/`
`models/causal_improved_sudormrf_v3.py`

Table 10: Comparison on separation performance of different models at low latency on the LRS2-2Mix dataset.

| Models | SI-SNRi (dB) | SDRi (dB) | Params (M) | MACs (G/s) |
|---|---|---|---|---|
| SuDoRM-RF 2.5x (causal) | 4.2 | 5.1 | 6.4 | 10.1 |
| A-FRCNN-16 (causal) | 8.9 | 9.5 | 6.1 | 125.3 |
| TDANet (causal) | 9.5 | 10.0 | 2.3 | 4.7 |

Table 11: Comparison on separation performance of models with different $L$'s on the LRS2-2Mix dataset.

| Name | SI-SNRi | SDRi | Params (M) | MACs (G/s) | Inference Time (s) |
|---|---|---|---|---|---|
| | | | $L = 4$ ms | | |
| A-FRCNN-16 | 12.5 | 12.8 | 6.2 | 37.3 | 1.32 |
| Sepformer | 12.0 | 12.3 | 26.0 | 23.5 | 1.89 |
| TDANet | 13.2 | 13.5 | 2.3 | 4.7 | 0.79 |
| | | | $L = 2$ ms | | |
| A-FRCNN-16 | 12.9 | 13.2 | 6.2 | 61.5 | 2.43 |
| Sepformer | 12.6 | 12.9 | 26.0 | 49.0 | 3.13 |
| TDANet | 14.2 | 14.5 | 2.3 | 9.1 | 1.78 |
| | | | $L = 1$ ms | | |
| A-FRCNN-16 | 13.0 | 13.3 | 6.1 | 125.3 | 5.32 |
| Sepformer | 13.5 | 13.8 | 26.0 | 86.9 | 7.55 |
| TDANet | 14.9 | 15.2 | 2.3 | 18.1 | 2.42 |

