# OpenReview forum: "An efficient encoder-decoder architecture with top-down attention for speech separation"
_ICLR.cc/2023/Conference — ICLR 2023 poster_

### Official Review · Reviewer_88kn · 2022-10-23

**Confidence:** 4
**Clarity, Quality, Novelty And Reproducibility:** see in the review
**Correctness:** 3
**Technical Novelty And Significance:** 2
**Empirical Novelty And Significance:** 2
**Recommendation:** 5

**Strength And Weaknesses:**

Experimental results show that although this model is small it gets competitive results to SOTA models which are larger less efficient. Furthermore, they show that each part of the architecture is essential for the best results

**Summary Of The Paper:**

In this paper, a lightweight blind source separation model (TDANet) is presented. The model is inspired from the brain’s top-down attention architecture.   The separation is carried out using an embedder, a separation and a decoder (with the  same parameters for all speakers) on the time samples.
The model is consisted with down and up sample units, global attention, and transformer to output acoustic features on this information and LA layers in the decoder.


**Summary Of The Review:**

Major concerns :

The bio-inspiration in this paper is emphasized many times although it is not clear at all to the reader. A short description can help understand the motivation for this approach.

The main idea of this paper is to build an efficient model for speech separation. The minimum number of parameters in the proposed method is 2.3 M. It is compared to other method with more parameters. Interestingly, the SuDoRM-RF0.5x and SuDoRM-RF0.25x with less parameters are not mentioned – please explain.

B equals to 16, it is not clear if this is considered in the complexity measurement.

The authors chose L=4ms and only 3 1d conv layers. The other models are using the same parameters ? if not, it is interesting to understand the contribution of these parameters to the method efficiency and performance.

The new LRS2-2mix dataset – It seems that the random method can chose for the same scenario 2 reverberated signals from different enclosures. Please elaborate on this.

Minor concerns :

To make it easier to follow, I think that it will be better to add the parameters names (G_m,F_i for instance) to the architecture figure.

In the experimental study, only the improvement is reported. To better understand the performance of the model, the noisy signal should also be mentioned as a reference. For instance, if the noisy input’ SI-SDR is -10, 10 dB improvement does not improve the intelligibility so much.

The SI-SDR measurement reported in the training method (eq. 8,9) is different from the original paper (I guess it is only a typo but if not, we cannot compare the results to the other methods)

The Sepformer model on the wham! Is reported in their paper with sisdri=16.4 while in this paper it is only 14.4 – please check

It is not clear what is the frequency sample used (8KHz/16KHz ?)

---

> ### Author Response · Authors · 2022-11-18
> **Answer to Reviewer 88kn (1/3)**
>
> Thank you very much for your valuable feedback. We have revised the manuscript and added some experiments to demonstrate the superiority of our model. We would also like to make some comments below to address your specific questions:
>
> ***Q1: The bio-inspiration in this paper is emphasized many times although it is unclear to the reader. A short description can help understand the motivation for this approach.***
>
> A1: The top-down attention in TDANet is inspired by the structure of the human brain. Due to length constraints, we will briefly describe it in the third paragraph of the introduction section. The detailed contents are shown below.
>
> The brain's sensory system is hierarchically organized, allowing sensory information (e.g., auditory information) to rise through a series of brain regions before reaching the higher sensory cortex (e.g., higher auditory cortex) [1].  Encoder-decoder speech separation networks (e.g., SuDORM-RF [2] and A-FRCNN [3]) contain a large number of top-down, bottom-up, and lateral connections, similar to the brain's hierarchical structure for processing sensory information.
>
> From a neuroscience perspective, mimicking the biological neural network architecture is a promising way to find a simple and efficient speech separation proposal. However, the aforementioned encoder-decoder architectures mainly simulated the interaction between lower (e.g., A1) and higher (e.g., A2) functional areas in primates, neglecting the role of higher cortices in speech signal processing. Moreover, selective attention is the process that allows individuals to select and focus on specific inputs for further processing while suppressing irrelevant or distracting information [4].
>
> In this paper, we focus on top-down attention, which is the foundation of the ability to focus on task-relevant stimuli and ignore irrelevant distractions [5-6]. Specifically, top-down modulation supports selective attention primarily by enhancing and inhibiting cortical sensory responses to different sensory information [7-8]. In addition, in intracranial recordings, some researchers [9] have found that the neural representation of sensory information in the auditory system is also modulated by attention. Additionally, the selective attention plays a critical role in tasks where task-independent signals or background noise are distinguished [9-13]. It is widely believed that this ability to distinguish between different audios is related to selective attention, a psychological process that allows individuals to focus on a specified target.
>
> However, the way how attentional mechanisms affect the processing of complex auditory scenes is largely unknown [14]. In this paper, we focus on designing neural networks for speech separation tasks by mimicking top-down attention in a hierarchical structure to achieve good results with high efficiency.
>
> ***Q2: The main idea of this paper is to build an efficient model for speech separation. The minimum number of parameters in the proposed method is 2.3 M. It is compared to other method with more parameters. Interestingly, the SuDoRM-RF0.5x and SuDoRM-RF0.25x with less parameters are not mentioned – please explain.***
>
> A2: We compared with larger versions of SuDoRM-RF (1.0x and 2.5x; Tables 2 and 3) instead of SuDoRM-RF0.5x and SuDoRM-RF0.25x only because the performances of these smaller models are too far below that of TDANet, as shown in the following tables. Though they are faster than TDANet, their accuracies are too low to make them practical solutions to speech separation. If you think it is necessary to include them in the final paper, we’ll do that.
>
>
> |Model|LRS2-2Mix||Libri2Mix||WHAM!||Params|
> |:----|:----|:----|:----|:----|:----|:----|:----|
> | |SI-SNRi|SDRi|SI-SNRi|SDRi|SI-SNRi|SDRi| |
> |SuDoRM-RF 0.25x|9.8|10.2|10.8|11.3|10.4|10.8|0.8|
> |SuDoRM-RF 0.5x|10.5|10.9|12.2|12.6|11.8|12.2|1.4|
> |TDANet|13.2|13.5|16.9|17.4|14.8|15.0|2.3|
>
> |Model|F GPU RTF (ms)|B GPU RTF (ms)|CPU RFT (s)|MACs (G/s)|
> |:----|:----|:----|:----|:----|
> |SuDoRM-RF 0.25x|7.65|33.65|0.25|1.9|
> |SuDoRM-RF 0.5x|9.21|72.63|0.35|2.8|
> |TDANet|11.76|86.56|0.79|4.7|
>
> ***Q3: B equals to 16, it is not clear if this is considered in the complexity measurement.***
>
> A3: We did take B=16 into account when measuring the model complexity. In other words, we calculated the complexity after we unfold the model 16 times.

---

> > ### Author Response · Authors · 2022-11-18
> > **Answer to Reviewer 88kn (2/3)**
> >
> > ***Q4: The authors chose L=4ms and only 3 1d conv layers. The other models are using the same parameters? if not, it is interesting to understand the contribution of these parameters to the method efficiency and performance.***
> >
> > A4: For the hyperparameter L, different models have different settings. We compare the results of the SOTA model Sepformer and lightweight model A-FRCNN with different values of L. It is seen from the following table that, for any model, when L is larger, the inference speed is faster, but the separation performance goes down. In addition, for any L value, our proposed TDANet achieves the best results with the highest efficiency. We have added this table into the Supplementary Materials.
> >
> > | Name       | SI-SNRi | SDRi | Params (M) | MACs (G/s) | Inference Time (s) |
> > |------------|---------|------|------------|------------|--------------------|
> > | ***L=4 ms***     |         |      |            |            |                    |
> > | A-FRCNN-16 | 12.5    | 12.8 | 6.2        | 37.3       | 1.32               |
> > | Sepformer  | 12.0    | 12.3 | 26.0       | 23.5       | 1.89               |
> > | TDANet     | 13.2    | 13.5 | 2.3        | 4.7        | 0.79               |
> > | ***L=2 ms***     |         |      |            |            |                    |
> > | A-FRCNN-16 | 12.9    | 13.2 | 6.2        | 61.5       | 2.43               |
> > | Sepformer  | 12.6    | 12.9 | 26.0       | 49.0       | 3.13               |
> > | TDANet     | 14.2    | 14.5 | 2.3        | 9.1        | 1.78               |
> > | ***L=1 ms***     |         |      |            |            |                    |
> > | A-FRCNN-16 | 13.0    | 13.3 | 6.1        | 125.3      | 5.32               |
> > | Sepformer  | 13.5    | 13.8 | 26.0       | 86.9       | 7.55               |
> > | TDANet     | 14.9    | 15.2 | 2.3        | 18.1       | 2.42               |
> >
> > We are unsure what you are referring to with "only 3 1d conv layer". We have two guesses:
> >
> > 1. Figure 1 shows that TDANet has three DWConv, which is set to S=3 to draw the graph to look concise. In Section 4.2, we illustrated S=4, meaning down-sampling for four times. This hyperparameter is consistent with A-FRCNN and SuDoRM-RF.
> >
> > 2. FFN layer in the Transformer layer, which contains three 1D convolutional layers. nn.TransformerEncoderLayer is also used in Sepformer, which has an FFN layer containing two linear layers. The number of FFN parameters in TDANet is 1.05M, and the complexity is 1.08 GMac/s, while the number of FFN parameters in Sepformer is 0.5M, and the complexity is 3.68 GMac/s.  Therefore, we believe that the inference speed of TDANet's FFN should be theoretically faster than Sepformer.
> >
> >
> > If you have time, would you please be more specific about this problem? Then we can address your concern more clearly.
> >
> > ***Q5: The new LRS2-2mix dataset – It seems that the random method can chose for the same scenario 2 reverberated signals from different enclosures. Please elaborate on this.***
> >
> > A5: The LRS2 corpus is organized in a way such that data of the same scene are stored in the same folder. When we randomly selected data to generate mixture audios, we avoided selecting speech data from the same folder so as to avoid getting mixture audios from the same scene. This generation approach follows the general principle of dataset generation in this domain. We have updated the manuscript on LRS2-2Mix data generation.
> >
> > ***Q6: To make it easier to follow, I think that it will be better to add the parameters names (G_m,F_i for instance) to the architecture figure.***
> >
> > A6: Thanks for the suggestion. We have added symbols to Figure 1 to promote a better understanding of the model structure.
> >
> > ***Q7: In the experimental study, only the improvement is reported. To better understand the performance of the model, the noisy signal should also be mentioned as a reference. For instance, if the noisy input’ SI-SDR is -10, 10 dB improvement does not improve the intelligibility so much.***
> >
> > A7: We think you want to see the results with the evaluation metric SI-SNR instead of SI-SNRi and SDRi. The following table shows SI-SNR of different models. Clearly, TDANet achieves better results than other models measured by this metric.
> >
> > | Model               | LRS2-2Mix | Libri2Mix | WHAM!  |
> > |---------------------|-----------|-----------|--------|
> > |                     | SI-SNR    | SI-SNR    | SI-SNR |
> > | BLSTM-TasNet        | 6.1       | 5.9       | 5.3    |
> > | Conv-TasNet         | 10.6      | 10.2      | 8.2    |
> > | SuDoRM-RF 1.0x      | 11.0      | 11.5      | 8.4    |
> > | SuDoRM-RF 2.5x      | 11.3      | 12.0      | 9.2    |
> > | DualPathRNN         | 12.7      | 12.1      | 9.2    |
> > | DPTNet              | 13.3      | 14.7      | 10.4   |
> > | A-FRCNN-16          | 13.0      | 14.7      | 10.0   |
> > | Sepformer           | 13.5      | 14.5      | 9.9    |
> > | TDANet (ours)       | 13.2      | 14.9      | 10.3   |
> > | TDANet Large (ours) | 14.2      | 15.4      | 10.7   |

---

> > > ### Author Response · Authors · 2022-11-18
> > > **Answer to Reviewer 88kn (3/3)**
> > >
> > > We reported SI-SNRi and SDRi in the paper because they are commonly used metrics in speech separation [2, 3, 15-19]. For your reference, the noise of Libri2Mix is -2.0 dB (see Table 4 in [20]) and the noise of WHAM! is -4.5 dB (see Table 2 in [21]). There are no clean target audios in LRS2, so the results with and without improvement are the same in calculating the evaluation metrics.
> > >
> > > ***Q8: The SI-SDR measurement reported in the training method (eq. 8,9) is different from the original paper (I guess it is only a typo but if not, we cannot compare the results to the other methods)***
> > >
> > > A8: It is indeed a typo. We missed $x_i$ in equation (8). Thanks for pointing it out. We apologize that we wrote the wrong formula for SI-SNR in the training method. We modified it to the following formula.
> > >
> > > $$A_{target}=\frac{\langle\bar{x}_i, x_i\rangle x_i}{||x_i||_2^2}.$$
> > >
> > > ***Q9: The Sepformer model on the wham! Is reported in their paper with sisdri=16.4 while in this paper it is only 14.4 – please check***
> > >
> > > A9: We would like to clarify that 16.4 reported in the original paper [22] was obtained after data augmentation, while all results of the compared models reported in our paper were obtained without data augmentation consistent with the settings in their original papers.
> > > The results listed in the SpeechBrain (https://github.com/speechbrain/speechbrain/tree/develop/recipes/WHAMandWHAMR/separation) show that data augmentation significantly impacts the performance (SI-SNRi is improved by 2.3 dB in the WHAMR! dataset). We asked the authors via email whether to publish the results of Sepformer without data enhancement, and the authors replied that they would not publish them for now. In addition, Sepformer's results were obtained by training with the code provided by SpeechBrain, so we believe the results we obtained for Sepformer are reliable and accurate.
> > >
> > > ***Q10: It is not clear what is the frequency sample used (8KHz/16KHz ?)***
> > >
> > > A10: The sampling rate of the LRS2-2Mix dataset is 16 kHz as described in Section 4.1, and the sampling rate of the Libri2Mix and WHAM! datasets are 8 kHz. These are public datasets and hence not specified in the paper. We have added the sampling rates of the Libri2Mix and WHAM! datasets into the manuscript.

---

> > > > ### Author Response · Authors · 2022-11-18
> > > > **Reference**
> > > >
> > > > [1] Rauschecker J P, Scott S K. Maps and streams in the auditory cortex: nonhuman primates illuminate human speech processing[J]. Nature neuroscience, 2009, 12(6): 718-724.
> > > >
> > > > [2] Tzinis E, Wang Z, Smaragdis P. Sudo rm-rf: Efficient networks for universal audio source separation[C], MLSP 2020.
> > > >
> > > > [3] Hu X, Li K, Zhang W, et al. Speech separation using an asynchronous fully recurrent convolutional neural network[C], NeurIPS 2021.
> > > >
> > > > [4] Mendoza L. Audiology: The Fundamentals[J]. Ear and Hearing, 1995, 16(4): 433-434.
> > > >
> > > > [5] Reynolds J H, Chelazzi L. Attentional modulation of visual processing[J]. Annual review of neuroscience, 2004, 27(1): 611-647.
> > > >
> > > > [6] Fries P. Neuronal gamma-band synchronization as a fundamental process in cortical computation[J]. Annual review of neuroscience, 2009, 32(1): 209-224.
> > > >
> > > > [7] Gazzaley A, Cooney J W, McEvoy K, et al. Top-down enhancement and suppression of the magnitude and speed of neural activity[J]. Journal of cognitive neuroscience, 2005, 17(3): 507-517.
> > > >
> > > > [8] Johnson J A, Zatorre R J. Attention to simultaneous unrelated auditory and visual events: behavioral and neural correlates[J]. Cerebral cortex, 2005, 15(10): 1609-1620.
> > > >
> > > > [9] Mesgarani N, Chang E F. Selective cortical representation of attended speaker in multi-talker speech perception[J]. Nature, 2012, 485(7397): 233-236.
> > > >
> > > > [10] Elhilali M, Xiang J, Shamma S A, et al. Interaction between attention and bottom-up saliency mediates the representation of foreground and background in an auditory scene[J]. PLoS biology, 2009, 7(6): e1000129.
> > > >
> > > > [11] Power A J, Foxe J J, Forde E J, et al. At what time is the cocktail party? A late locus of selective attention to natural speech[J]. European Journal of Neuroscience, 2012, 35(9): 1497-1503.
> > > >
> > > > [12] Rimmele J M, Golumbic E Z, Schröger E, et al. The effects of selective attention and speech acoustics on neural speech-tracking in a multi-talker scene[J]. Cortex, 2015, 68: 144-154.
> > > >
> > > > [13] Golumbic E M Z, Ding N, Bickel S, et al. Mechanisms underlying selective neuronal tracking of attended speech at a “cocktail party”[J]. Neuron, 2013, 77(5): 980-991.
> > > >
> > > > [14] Elhilali M, Xiang J, Shamma S A, et al. Interaction between attention and bottom-up saliency mediates the representation of foreground and background in an auditory scene[J]. PLoS biology, 2009, 7(6): e1000129.
> > > >
> > > > [15] Luo Y, Chen Z, Yoshioka T. Dual-path rnn: efficient long sequence modeling for time-domain single-channel speech separation[C], ICASSP 2020.
> > > >
> > > > [16] Subakan C, Ravanelli M, Cornell S, et al. Attention is all you need in speech separation[C], ICASSP, 2021.
> > > >
> > > > [17] Nachmani, Eliya, Yossi Adi, and Lior Wolf. Voice separation with an unknown number of multiple speakers[C], ICML 2020.
> > > >
> > > > [18] Lam, Max WY, et al. Effective Low-Cost Time-Domain Audio Separation Using Globally Attentive Locally Recurrent Networks[C], SLT 2021.
> > > >
> > > > [19] Subakan, Cem, et al. Resource-efficient separation transformer[J]. arXiv preprint arXiv:2206.09507, 2022.
> > > >
> > > > [20] Cosentino J, Pariente M, Cornell S, et al. Librimix: An open-source dataset for generalizable speech separation[J]. arXiv preprint arXiv:2005.11262, 2020.
> > > >
> > > > [21] Wichern G, Antognini J, Flynn M, et al. WHAM!: Extending Speech Separation to Noisy Environments[C], Interspeech 2019.
> > > >
> > > > [22] Subakan C, Ravanelli M, Cornell S, et al. On Using Transformers for Speech-Separation[J]. arXiv preprint arXiv:2202.02884, 2022.

---

> > > > ### Comment · Reviewer_88kn · 2022-11-29
> > > > **Regarding Q9**
> > > >
> > > > The proposed model gained almost the same results as of the Sepformer with the data augmentation.
> > > > While reducing the data augmentation your model outperforms the Sepformer.
> > > > Although your model is lighter, it is still not clear why you do not use the Sepformer with the data augmentation and use the same augmentation method to your model.

---

> > > > > ### Author Response · Authors · 2022-11-30
> > > > > **Further Response to Reviewer 88kn (Q9)**
> > > > >
> > > > > Dear Reviewer 88kn,
> > > > >
> > > > > We have the following three responses regarding whether to use data augmentation consistent with Sepformer.
> > > > >
> > > > > First, we clarify again that almost all existing methods for speech separation obtain results ***without data augmentation***, consistent with the setup in their original papers (see A9).
> > > > >
> > > > > Second, the impact of data augmentation may be different for different methods. We tried to use the same data augmentation on the WHAM! dataset as Sepformer but only obtained marginal gains (see table below).
> > > > >
> > > > > |    Data   augmentation    |    SI-SNRi    |     SDRi    |
> > > > > |:-------------------------:|:-------------:|:-----------:|
> > > > > |              √            |      15.7     |     15.9    |
> > > > > |              x            |      15.2     |     15.4    |
> > > > >
> > > > > We investigated Sepformer's official code repository and found that another speech separation method SkiM [1] they reproduced using the same data augmentation as Sepformer yielded lower separation performance on the WSJ0-2Mix dataset than the performance described in the original paper without data augmentation.  See the table below, where the SkiM's results using data augmentation were provided by Speechbrain [https://drive.google.com/drive/folders/1XmvH4p7UkXT7kend0wbV2PYmdBIrG8Vz](https://drive.google.com/drive/folders/1XmvH4p7UkXT7kend0wbV2PYmdBIrG8Vz).
> > > > > , and the results without data augmentation are reported in [1].
> > > > >
> > > > > |     Data   augmentation    |     SI-SNRi    |     SDRi    |
> > > > > |:--------------------------:|:--------------:|:-----------:|
> > > > > |              √             |       18.1     |     18.3    |
> > > > > |              x             |       18.3     |     18.7    |
> > > > >
> > > > > Therefore, it seems that the impact of data augmentation may be different for different methods. An in-depth investigation on the impact of data augmentation is valuable, but that is outside the scope of this work. Due to the complexity of the effect of data augmentation over performance, introducing data augmentation could be more of an interference than a help to illustrate our point. We are more concerned with the modeling capabilities and performance of the model with raw data.
> > > > >
> > > > > As emphasized in the title, this work mainly aims at efficiency. Compared to Sepformer, TDANet has an approximately 11 times smaller number of parameters, about 18 times smaller MAC, about 9 times smaller inference time on CPU, about five times smaller inference time on the GPU, and approximately twice smaller training time on GPU.
> > > > >
> > > > > Kind regards,
> > > > >
> > > > > The authors
> > > > >
> > > > > ***References***
> > > > >
> > > > > [1] Li C, Yang L, Wang W, et al. SkiM: Skipping Memory LSTM for Low-Latency Real-Time Continuous Speech Separation[C], ICASSP 2022.

---

> > > ### Comment · Reviewer_88kn · 2022-11-29
> > > **Regarding Q5+Q7**
> > >
> > > Regarding Q5 - If I understand you correctly,  for each mixture you mixed 2 reverberated signals recorded in 2 different rooms. If this is the case it is not so realistic. This might shade some light on the high performance in the LRS2-2Mix column.
> > > In real life, the mixture is recorded in the same scene and the simulated dataset should keep this property.

---

> > > > ### Author Response · Authors · 2022-11-30
> > > > **Further Response to Reviewer 88kn (Q5+Q7)**
> > > >
> > > > Dear Reviewer 88kn,
> > > >
> > > > Concerning your question about using LRS2 data to construct the dataset, our answers are as follows.
> > > > 1. Indeed, LRS2 can only be mixed using audio from different scenes, since it does not contain the speech of different speakers from the same scenes (it is difficult to collect data in the same scenes).
> > > >
> > > > 2. We understand your concerns about the LRS2-2Mix dataset. However, almost all existing multimodal speech separation methods ***[1-6]*** use this dataset for training and testing, mixed similarly as presented in our paper. And they are also able to obtain better generalizations in real scenarios.
> > > >
> > > > Kind regards,
> > > >
> > > > The authors
> > > >
> > > > ***References***
> > > >
> > > > [1] Gao R, Grauman K. Visualvoice: Audio-visual speech separation with cross-modal consistency[C], CVPR 2021.
> > > >
> > > > [2] Gu R, Zhang S X, Xu Y, et al. Multi-modal multi-channel target speech separation[J]. IEEE Journal of Selected Topics in Signal Processing, 2020.
> > > >
> > > > [3] Wu J, Xu Y, Zhang S X, et al. Time domain audio visual speech separation[C], ASRU 2019.
> > > >
> > > > [4] Lee J, Chung S W, Kim S, et al. Looking into your speech: Learning cross-modal affinity for audio-visual speech separation[C], CVPR 2021.
> > > >
> > > > [5] Li G, Yu J, Deng J, et al. Audio-visual multi-channel speech separation, dereverberation and recognition[C], ICASSP 2022.
> > > >
> > > > [6] Xiong J, Zhang P, Xie L, et al. Audio-visual speech separation based on joint feature representation with cross-modal attention[J]. arXiv preprint arXiv:2203.02655, 2022.

---

> > ### Comment · Reviewer_88kn · 2022-11-29
> > **Regarding Q1+Q2**
> >
> > Thank you for your detailed answer.
> > Regarding Q1 - It is much easier to understand the bio-inspirational now.
> >
> > Regarding Q2 - I think it should be added for a fair comparison, although I still have concerns about the numbers (see my next note).

---

> ### Author Response · Authors · 2022-11-30
> **Thank you very much for your further comments!**
>
> Dear Reviewer 88kn:
>
> Thank you very much for your further comments. They are very valuable in improving our work. If you have any additional questions, please let us know.
>
> We can understand your concerns about our newly proposed dataset. However, our approach has also been validated on two other public datasets (Libri2Mix and WHAM!). The results show that TDANet is also competitive in terms of performance while outperforming existing methods in terms of efficiency. It is worth noting that we also experimented in low latency (causal) mode (see the response to reviewer tcwu's A2) and found that TDANet still achieves better separation performance, demonstrating the ability to process in real-time with practical usage. We would appreciate if you could reevaluate our paper.
>
> Thank you.
>
> Authors.

---

> > ### Comment · Reviewer_88kn · 2022-12-01
> > **I have updated my recommendation**
> >
> > Thank you for your response.
> > I updated my recommendation.
> > It is evident that the proposed method is beneficial in terms of efficiency.
> > I still have problems with the tested datasets. One is problematic as you understand, and the other 2 do not take the acoustic scenario into account.

---

> > > ### Author Response · Authors · 2022-12-01
> > > **Thank you for your recommendation**
> > >
> > > Dear Reviewer 88kn:
> > >
> > > Thanks for increasing the score!
> > >
> > > We are not sure what you mean by mentioning that the other two datasets (WHAM! and LibriMix) do not consider acoustic scenarios. Both datasets are created by mixing clean speeches then adding noise to the mixtures, and they can be regarded as recorded from the same environment. And they have been widely used in the speech separation community. Creating more reasonable datasets is of course very important, but that will diverge the focus of this work. We appreciate your suggestion and will study it in future.
> > >
> > > Thank you!
> > >
> > > Authors

---

### Official Review · Reviewer_tcwu · 2022-10-24

**Confidence:** 4
**Correctness:** 3
**Technical Novelty And Significance:** 3
**Empirical Novelty And Significance:** 3
**Recommendation:** 6

**Clarity, Quality, Novelty And Reproducibility:**

The paper is clearly presented, with a full review of related research, the proposed idea is clarified. But there are a few flaws on writing, like in page 6, sec. 4, dataset name should be WHAM! instead of WAHM!. The authors should ensure the correctness of paper writing to avoid misunderstanding.

For the task of speaker separation, this work shows a worthwhile trial of approaching the target on a limited memory cost, which is less discussed in previous methods. This is good. And the use of top-down processes for the goal makes sense, which is surely inspired by related research on neurosciences. But a more thorough discussion/comparison of possible top-down attention options would be more persuasive and stronger for supporting the authors’ claim. Especially when there are abundant corpus of literature about top-down bottom-up attention in multiple AI fields, to name a few [1-3]. Principles can be learned from these works for a comprehensive assessment of the top-down bottom-up architecture.

Details about the proposed methods are sufficiently introduced by the paper, which is friendly to the readers for reproduction.

[1] Pinto Y, van der Leij A R, Sligte I G, et al. Bottom-up and top-down attention are independent[J]. Journal of vision, 2013, 13(3): 16-16.

[2] Jaiswal S, Fernando B, Tan C. TDAM: Top-Down Attention Module for Contextually Guided Feature Selection in CNNs[J].

[3] Wu T, Zhu S C. A numerical study of the bottom-up and top-down inference processes in and-or graphs[J]. International journal of computer vision, 2011, 93(2): 226-252.

**Strength And Weaknesses:**

Strengths: the authors work towards realistic app scenario for the speech separation task, which is a good point. The top-down attention is another good point which can be inspired by human brain knowledge.

Weaknesses: Beyond model size, the computational complexity is also unavoidable for practical uses on low-resource devices. It’s quite curious how the model behaves on the low latency (causal) mode, whether it is potential for a real-time processing speed. The design of TDNet also needs more discussions, seems the current one uses more existing architectures which can be interpreted as a top-down process. It would be better to analyze what the model learns in such a top-down process while previous methods did not. Or more analysis should be provided to show the insightful contribution of such a top-down network design.

**Summary Of The Paper:**

This paper proposed a new top-down attention framework for the task of speech separation. The knowledge is from human brain mechanism. The authors show that their method can achieve better separation results with a rather smaller model compared with recent approaches.

**Summary Of The Review:**

This paper provides a good trial on applying top-down bottom-up processes with MHSA on the task of speaker separation. The motivation of reducing model complexity is aligned with top-down bottom-up mechanism in human perception. However, the paper merely focus on intuitive design of network modules, without insightful & thorough analysis of the proposed top-down attention solution, especially compared with many related trials in other application fields. It is necessary to add more analysis for a wider interest in the ICLR community instead of solving a specific downstream task of speaker separation. As an efficient solution, it would be better if causal version performance can be shown for the potential of real-time processing. Paper writing should be carefully examined to avoid mistakes.

Although there are several flaws as mentioned, this work still proposes a novel architecture based on a clear intuition and achieves SOTA on more challenging scenarios of speaker separation task. It would be inspiring of this work in the field of speech processing. Hence I would suggest a weak accept of this paper to ICLR.

---

> ### Author Response · Authors · 2022-11-18
> **Answer to Reviewer tcwu (1/2)**
>
> We would like to thank you for your very positive feedback. We are honored that you have found potential in our work. Please find below our answers to the main questions that you have raised:
>
> ***Q1: Beyond model size, the computational complexity is also unavoidable for practical uses on low-resource devices.***
>
> A1: We show the computational complexity of the different models in Table 3. From the table, it is seen that the MACs of our model are more than ten times smaller than the SOTA model (Sepformer), and the inference time is about nine times faster on CPU. Although we did not perform real tests on low-resource devices, we believe that TDANet's inference time and separation performance are at least faster and better than other existing methods on the same devices.
>
> ***Q2: It’s quite curious how the model behaves on the low latency (causal) mode, whether it is potential for a real-time processing speed.***
>
> A2: For the real-time problem, we tried to modify SuDoRM-RF, A-FRCNN and TDANet to the causal version. We replaced the standard convolutional layer in both models with causal convolutional layer to mask out future information (the code for this convolution operation is shown below, modified from SuDoRM-RF). We supplemented the experiments with SuDoRM-RF(causal), A-FRCNN-16 (causal) and TDANet (causal), and the results are shown in the table below. The results show that modifying the models to the causal versions degrade the separation performance to varying degrees. However, TDANet is still able to obtain better performance than SuDoRM-RF and A-FRCNN.
>
> |  Model | SI-SNRi  | SDRi  |  Params (M) | MACs (G/s)  |
> | :------------: | :------------: | :------------: | :------------: | :------------: |
> | SuDoRM-RF 2.5x| 11.3| 11.7| 6.4| 10.1|
> | SuDoRM-RF 2.5x (causal)| 4.2| 5.1| 6.4| 10.1|
> | A-FRCNN-16| 13.0| 13.3| 6.1| 125.3|
> | A-FRCNN-16 (causal)| 8.9| 9.5| 6.1| 125.3|
> | TDANet| 13.2| 13.5| 2.3| 4.7|
> | TDANet (causal)| 9.5| 10.0| 2.3| 4.7|
>
> Code is adapted from https://github.com/etzinis/sudo_rm_rf
>
> ```python
> class CausalConv1d(nn.Conv1d):
>     """Causal 1D Conv layer"""
>     def __init__(self, in_channels, out_channels, kernel_size,
>                  stride=1, padding=0,
>                  dilation=1, groups=1, bias=True):
>         nn.Conv1d.__init__(self, in_channels, out_channels,
>                            kernel_size, stride, padding, dilation,
>                            groups, bias)
>         self.causal_mask = torch.ones_like(self.weight)
>         if kernel_size >= 3:
>             future_samples = kernel_size // 2
>             self.causal_mask[..., -future_samples:] = 0.
>
>     def forward(self, x):
>         return nn.functional.conv1d(
>             x, self.get_weight(), self.bias,
>             self.stride, self.padding, self.dilation, self.groups)
> ```
>
> ***Q3: What are the advantages of top-down attention in the GA module and decoder?***
>
> A3: In the reconstruction process, the encoder and decoder information are not always useful, so we need an automatic method to modulate the features transmitted by the lateral and top-down connections. Table 1 shows that adding top-down attention to the GA module and decoder can improve the speech separation performance.
>
> In the GA module, different from UNet, we use top-down attention to filter features from lateral connections before passing them to the decoder. This process is equivalent to a modulation, controlling the redundancy of information passed to the decoder. We verify the effect of global top-down attention again on SuDoRM-RF. We found that adding top-down attention to the SuDoRM-RF can also improve the separation performance, as shown in the following table.
>
> |  Model | SI-SNRi  | SDRi  |  Params (M) | MACs (G/s)  |
> | :------------: | :------------: | :------------: | :------------: | :------------: |
> | SuDoRM-RF| 11.0| 11.4| 2.7| 4.6|
> | SuDoRM-RF + Top-down ATT| 11.9| 12.3| 2.7| 4.6|
>
> In the LA layer of decoder, we use the features from the adjacent layer to obtain a set of parameters ($\rho$,$\tau$) as local top-down attention to perform affine transformation on features from the current layer to reconstruct fine-grained features (since we need to reconstruct the separated audios, fine-grained features are required). We add LA layers to the decoder of SuDoRM-RF to verify the importance of the LA layer. We found that adding LA layers can also improve the separation performance, as shown in the following table.
>
> |  Model | SI-SNRi  | SDRi  |  Params (M) | MACs (G/s)  |
> | :------------: | :------------: | :------------: | :------------: | :------------: |
> | SuDoRM-RF| 11.0| 11.4| 2.7| 4.6|
> | SuDoRM-RF+LA| 11.7| 12.0| 3.3| 4.9|

---

> > ### Author Response · Authors · 2022-11-18
> > **Answer to Reviewer tcwu (2/2)**
> >
> > ***Q4: The relationship between top-down attention in this paper and existing methods (not limited to speech separation).***
> >
> > A4: To the best of our knowledge, there is no method to use top-down attention for speech separation in the encoder-decoder architecture.
> >
> > Thus, we investigated the use of global top-down attention (similar to top-down attention in the GA module) in hierarchical models in other domains, e.g., image segmentation [1-2] and image fusion [3]. However, there are two differences between these methods and TDANet. First, TDANet uses the fused multi-scale features to obtain top-down attention, while they use different scale features to obtain the corresponding top-down attention. Second, these methods are too complex. For example, the semantically guided attention module in [2] has two refinement steps, while we only need one step by element-wise production.
> >
> > Furthermore, we surveyed local top-down attention approaches (similar to the LA layer) in other domains, such as SENet [4], Highway network [5] and GRCNN [6]. The Highway network [5] introduces an attention mechanism to modulate this layer's input, which alleviates the problem of gradient disappearance to some extent. SENet [4] essentially enables the model to learn the weight coefficients of each channel as attention to dynamically modulate the output feature map from the convolutional layer. These methods introduce the attention mechanism (gate) to achieve filtering and integration of information flow and obtain better performance.
> >
> > ***References***
> >
> > [1] Chen L C, Yang Y, Wang J, et al. Attention to scale: Scale-aware semantic image segmentation[C], CVPR 2016.
> >
> > [2] Sinha A, Dolz J. Multi-scale self-guided attention for medical image segmentation[J]. IEEE journal of biomedical and health informatics, 2020, 25(1): 121-130.
> >
> > [3] Li X, Chen H, Li Y, et al. MAFusion: Multiscale Attention Network for Infrared and Visible Image Fusion[J]. IEEE Transactions on Instrumentation and Measurement, 2022, 71: 1-16.
> >
> > [4] Hu J, Shen L, Sun G. Squeeze-and-excitation networks[C], CVPR 2018.
> >
> > [5] Srivastava R K, Greff K, Schmidhuber J. Highway networks[J]. arXiv preprint arXiv:1505.00387, 2015.
> >
> > [6] Wang J, Hu X. Convolutional neural networks with gated recurrent connections[J], TPAMI 2021.

---

> > > ### Comment · Reviewer_tcwu · 2022-12-01
> > > **Final comment**
> > >
> > > Thanks for the authors' response. My evaluation stays the same.

---

### Official Review · Reviewer_HQhT · 2022-10-25

**Confidence:** 4
**Correctness:** 3
**Technical Novelty And Significance:** 3
**Empirical Novelty And Significance:** 3
**Recommendation:** 6

**Clarity, Quality, Novelty And Reproducibility:**

Clarity is fine.
Novelty is acceptable.
Work is reproducible from the author's code.

**Strength And Weaknesses:**

Overall the paper is easy to follow, and the topic addressed is interesting.

- The proposed attention mechanism has similar ideas to the UNet autoencoder and other similar generative models in the area of empirical deep learning that combine evidence from multiple scales. The authors have not discussed such ideas in this paper.
- There is no theoretical discussion on the impact of individual modules and their interaction for overall performance gains. e.g., why adopt such a complicated pipeline?
Encoder: is Resnet style model with additional pooling layers at each layer. I see no advantage of this over a regular ResNet style architecture which also combines information from a previous block/layer operating at a different scale.
- The use of the transformer layer is also not clear. It seems like an unnecessary complication. Transformers are for sequence modeling, but I don't see such a requirement here. Also, the input positional encoding and training of a transformer using masking is very important for its effectiveness.




**Summary Of The Paper:**

The authors have proposed a deep learning-based framework for speech separation.
The main claim is the effectiveness of the top-down attention mechanism that works across multiple scales to recover the signal.

**Summary Of The Review:**

Please see my detailed comments.

---

> ### Author Response · Authors · 2022-11-18
> **Answer to Reviewer HQhT (1/2)**
>
> We would like to thank you for the time taken to review our paper and for your extensive comments. Your suggestions are very helpful, and we believe we can significantly improve the paper by making the respective adjustments. We have compiled the issues that you have pointed out and answer them as follows:
>
> ***Q1: The proposed attention mechanism has similar ideas to the UNet autoencoder and other similar generative models in the area of empirical deep learning that combine evidence from multiple scales. The authors have not discussed such ideas in this paper.***
>
> A1: In the paper, we mentioned SuDoRM-RF many times. It is a speech separation model that cascades multiple UNet and is also one of the baseline models we focus on for comparison. If we remove the GA module and LA layers from the basic building blocks of the proposed TDANet (see Table 1), it degenerates into a UNet.
>
> To the best of our knowledge, TDANet is the first to add top-down attention to the encoder-decoder architecture (UNet) in the speech separation domain. In the third paragraph of the related work section, we discuss existing methods use top-down attention mechanism to the top layer of the multilayer LSTM, which is different from the top-down attention modulated to multi-scale features proposed in our paper.
>
> Besides, we investigated the use of top-down attention in encoder-decoder architectures in other domains, e.g., image segmentation [1-2] and image fusion [3]. TDANet has three aspects different from [1-3]. First, the essential difference between TDANet and [1-3] is that TDANet is a recurrent neural network (RNN) because each basic building block of TDANet shares parameters, whereas they are feed-forward networks. Second, [1-3] use different scale features to obtain the corresponding top-down attention, while TDANet uses the fused multi-scale features to obtain top-down attention. Third, these methods are too complex. For example, the semantically guided attention module in [2] has two refinement steps, while we only need one step by element-wise production. We have added these discussions to our related work section.
>
> ***Q2: There is no theoretical discussion on the impact of individual modules and their interaction for overall performance gains. e.g., why adopt such a complicated pipeline? Encoder: is Resnet style model with additional pooling layers at each layer. I see no advantage of this over a regular ResNet style architecture which also combines information from a previous block/layer operating at a different scale.***
>
> A2: The following is discussions of different structures in the TDANet (Encoder, Global attention module (GA) and Decoder).
>
> ***Encoder***: Actually, we are not concerned with which model TDANet's Encoder uses to extract features ($F_{i}$) of different scales, because it is not the contribution we want to highlight. Our contribution is to propose the GA module and the Decoder for TDANet. The encoder is a down-sampling structure consisting of several layers containing one 1D convolutional layer, which is more straightforward than ResNet. Encoder may achieve better performance when replaced by ResNet separation, but it is not our focus. Note that the dense connections (additional pooling layers) are not involved in the information processing of Encoder. They only process different scale features as input to the GA module. See below for a discussion about these dense connections.
>
> ***Global attention (GA) module***: The GA module consists of a transformer layer and top-down attention. The transformer layer's input is the fused multi-scale features $G$ processed using dense connections. The dense connections only use the pooling layer without any parameters. When we remove the dense connections, TDANet's encoder becomes a typical UNet's encoder. We use dense connections for the following two reasons:
>
> (1) Dense connections are like DenseNet [4], boosting the backpropagation of gradients and making the network easier to train.
>
> (2) The features may lose some details during bottom-up routing. Using skip connections to project to the top layer enables more efficient use of multi-scale features.
>
> The reason for using transformer layer modeling is that this network is designed explicitly for sequence modeling compatible with the speech separation task (sequence task) and is also a standard attention model.
>
> Using top-down attention in modulating features at different scales is extremely simple in terms of implementation (we only need one step by element-wise production). We use top-down attention to modulate local features reducing information redundancy, allowing the network to focus more on task-relevant features and better guide the modeling process for sequences of different scales.

---

> > ### Author Response · Authors · 2022-11-18
> > **Answer to Reviewer HQhT (2/2)**
> >
> > ***Decoder***: After removing the gray box in Figure. 4, it becomes a decoder in a typical UNet. The LA layer is just two 1D depthwise convolutional layers and a Sigmoid function, with only ~0.01M parameters. Table 1 shows that the LA layers in the decoder improve the separation performance (1.8 dB gains). One possible reason is that the LA layer uses the neighboring layer features to learn a set of parameters to adaptively modulate the fused features of the current layer to reconstruct fine-grained features.
> >
> > This operation was unavailable in previous UNet-based models (SuDoRM-RF [5]). Therefore, we verify the importance of the LA layer again for the SuDoRM-RF model by adding the LA layer to the decoder of SuDoRM-RF and obtain the results in the following table. We can find that the LA layer can improve separation performance without increasing the number of parameters and computational cost.
> >
> > |  Model | SI-SNRi  | SDRi  |  Params (M) | MACs (G/s)  |
> > | :------------: | :------------: | :------------: | :------------: | :------------: |
> > |  SuDoRM-RF |  11.0 |  11.4 |  2.7 |  4.6 |
> > | SuDoRM-RF+ LA  |  11.7 |  12.0 | 3.3  |  4.9 |
> >
> > We have added these analyses to the paper to facilitate reader's understanding of the network structure of TDANet.
> >
> > ***Q3: The use of the transformer layer is also not clear. It seems like an unnecessary complication. Transformers are for sequence modeling, but I don't see such a requirement here. Also, the input positional encoding and training of a transformer using masking is very important for its effectiveness.***
> >
> > A3: Thanks to the reviewers for the question. One of the concepts we need to clarify is that speech separation is a sequence reconstruction task that separates speeches of several individual speakers from a mixture speech containing these speakers, so the transformer layer is well suited for this task. In addition, other existing approaches also used the transformer layer, such as Sepformer [7] and DPTNet [8]. We used the transformer layer because we want the model to learn the internal structure of speech and some semantic features for contextual modeling on speech sequences using the self-attention mechanism, which effectively builds long-term memory.
> >
> > In addition, for the position encoding and masking issue you mentioned, we used sine and cosine functions of different frequencies for position encoding (see Section 4.2) and did not use masking (because our task is to be able to see future information).
> >
> > ***References***
> >
> > [1] Chen L C, Yang Y, Wang J, et al. Attention to scale: Scale-aware semantic image segmentation[C], CVPR 2016.
> >
> > [2] Sinha A, Dolz J. Multi-scale self-guided attention for medical image segmentation[J]. IEEE journal of biomedical and health informatics, 2020, 25(1): 121-130.
> >
> > [3] Li X, Chen H, Li Y, et al. MAFusion: Multiscale Attention Network for Infrared and Visible Image Fusion[J]. IEEE Transactions on Instrumentation and Measurement, 2022, 71: 1-16.
> >
> > [4] Huang G, Liu Z, Van Der Maaten L, et al. Densely connected convolutional networks[C], CVPR 2017.
> >
> > [5] Tzinis E, Wang Z, Smaragdis P. Sudo rm-rf: Efficient networks for universal audio source separation[C], MLSP 2020.
> >
> > [6] Hu X, Li K, Zhang W, et al. Speech separation using an asynchronous fully recurrent convolutional neural network[C], NeurIPS 2021.
> >
> > [7] Subakan C, Ravanelli M, Cornell S, et al. Attention is all you need in speech separation[C], ICASSP 2021.
> >
> > [8] Chen J, Mao Q, Liu D. Dual-Path Transformer Network: Direct Context-Aware Modeling for End-to-End Monaural Speech Separation[C], Interspeech 2020.

---

> > ### Comment · Reviewer_HQhT · 2022-12-01
> > **Final comments**
> >
> > Thanks for responding and clarifying the points.
> > Overall my evaluation still remains unchanged.

---

### Author Response · Authors · 2022-11-18
**To all reviewers**

We would like to thank all the reviewers for the helpful comments and suggestions, and we sincerely appreciate the time and efforts the reviewers put on this manuscript. Please find the point-to-point response to all the comments below.

We have updated the paper with some updates based on your comments (the updated sections are highlighted in red).

---

### Author Response · Authors · 2022-11-22
**Hope the reviewers will take note of our response**

Dear Reviewers,

After submitting the initial comments, we incorporated your feedback into a revised version of our paper, performed some additional experiments as you requested, and wrote a response to address your main concerns.

We hope to interact with you during the discussion and potentially further improve the quality of our paper.

Thank you very much in advance.

Kind regards,

Authors

---

### Decision · Program_Chairs · 2023-01-20

**Decision:**

Accept: poster

**Justification For Why Not Higher Score:**

The paper is a little speech-specific. it will be great that the work/idea can be generalized to more domains.

**Justification For Why Not Lower Score:**

The authors actively addressed all review comments by adding new experiment results. Overall the paper quality is good. The proposed method is effective. Compared to other existing methods, the proposed one has much less computational cost. Note that the database challenge from one reviewer may not be an issue because the authors used standard databases on the task.

**Metareview: Summary, Strengths And Weaknesses:**

Paper Summary:

The authors propose a bio-inspired method neural network architecture for speech separation, where a top-down attention and a local attention module are introduced to the U-Net based separation network. The proposed method has shown a better performance but with a significant less computation cost, than several recent separation networks.

Novelty:

Reviewers have raised questions on the novelty of the paper, compared with baseline UNET. As the author pointed out, the main difference, i.e. the major contribution of the paper lies in the top down attention mechanism to speech separation. In additional experiment part, the authors further verifies the effectiveness of the top-down attention(TDA) by incorporating the SuDoRM-RF network with the TDA module. Together with the result for the result of the proposed model, method seems convincing.


Effectiveness:

Some reviewers raised the question for causal processing, which is answered by additional experiment results added to the appendix.
A question on the comparison with baseline model was raised, where the baseline models have different hyper-parameter setting in window length. This question was answered by additional experiment results.

Clarity:

The clarity of the paper is good. Some reviewers raised comments on insufficient explanation in bio-logical neural processing, which is later improved by the modification of the paper in corresponding section.

After additional experiment results and updates in description. The novelty, effectiveness and clarity seem good for publication in ICLR 2023


**Note From Pc:**

if the above contains the word "oral" or "spotlight" please see: "oral" presentation means -> notable-top-5% and "spotlight" means -> notable-top-25%. As stated in our emails, we are disassociating presentation type from AC recommendations